# Unification Theories: New Results and Examples

**Florin F. Nichita**

Simion Stoilow Institute of Mathematics of the Romanian Academy 21 Calea Grivitei Street, 010702 Bucharest, Romania; florin.nichita@imar.ro; Tel.: +40-0-21-319-6506; Fax: +40-0-21-319-6505

**Abstract:** This paper is a continuation of a previous article that appeared in AXIOMS in 2018. A Euler's formula for hyperbolic functions is considered a consequence of a unifying point of view. Then, the unification of Jordan, Lie, and associative algebras is revisited. We also explain that derivations and co-derivations can be unified. Finally, we consider a "modified" Yang–Baxter type equation, which unifies several problems in mathematics.

**Keywords:** Euler's formula; hyperbolic functions; Yang–Baxter equation; Jordan algebras; Lie algebras; associative algebras; UJLA structures; (co)derivation

## 1. Introduction

Voted the most famous formula by undergraduate students, the Euler's identity states that $e^{\pi i} + 1 = 0$. This is a particular case of the Euler's–De Moivre formula:

$$\cos x + i \sin x = e^{ix} \quad \forall x \in \mathbb{R}, \tag{1}$$

and, for hyperbolic functions, we have an analogous formula:

$$\cosh x + J \sinh x = e^{xJ} \quad \forall x \in \mathbb{C}, \tag{2}$$

where we consider the matrices

$$J = \begin{pmatrix} 0 & 0 & 0 & 1 \\ 0 & 0 & 1 & 0 \\ 0 & 1 & 0 & 0 \\ 1 & 0 & 0 & 0 \end{pmatrix} \tag{3}$$

$$I = \begin{pmatrix} 1 & 0 & 0 & 0 \\ 0 & 1 & 0 & 0 \\ 0 & 0 & 1 & 0 \\ 0 & 0 & 0 & 1 \end{pmatrix} \tag{4}$$

$$I' = \begin{pmatrix} 1 & 0 \\ 0 & 1 \end{pmatrix} . \tag{5}$$

In fact, $R(x) = \cosh(x)I + \sinh(x)J = \cosh x + J \sinh x = e^{xJ}$ also satisfies the equation

$$(R \otimes I')(x) \circ (I' \otimes R)(x+y) \circ (R \otimes I')(y) = (I' \otimes R)(y) \circ (R \otimes I')(x+y) \circ (I' \otimes R)(x) \tag{6}$$

called the colored Yang–Baxter equation. This fact follows easily from $J^{12} \circ J^{23} = J^{23} \circ J^{12}$ and $x J^{12} + (x+y) J^{23} + y J^{12} = y J^{23} + (x+y) J^{12} + x J^{23}$, and it shows that the formulas (1) and (2) are related.

While we do not know a remarkable identity related to (2), let us recall an interesting inequality from a previous paper: $|e^i - \pi| > e$. There is an open problem to find the matrix version of this inequality.

The above analysis is a consequence of a unifying point of view from previous papers ([1,2]).

In the remainder of this paper, we first consider the unification of the Jordan, Lie, and associative algebras. In Section 3, we explain that derivations and co-derivations can be unified. We suggest applications in differential geometry. Finally, we consider a "modified" Yang–Baxter equation which unifies the problem of the three matrices, generalized eigenvalue problems, and the Yang–Baxter matrix equation. There are several versions of the Yang–Baxter equation (see, for example, [3,4]) presented throughout this paper.

We work over the field $k$, and the tensor products are defined over $k$.

## 2. Weak Ujla Structures, Dual Structures, Unification

**Definition 1.** *(Ref. [5]) Given a vector space V, with a linear map $\eta : V \otimes V \to V$, $\eta(a \otimes b) = ab$, the couple $(V, \eta)$ is called a "weak UJLA structure" if the product $ab = \eta(a \otimes b)$ satisfies the identity*

$$(ab)c + (bc)a + (ca)b = a(bc) + b(ca) + c(ab) \quad \forall\, a, b, c \in V. \tag{7}$$

**Definition 2.** *Given a vector space V, with a linear map $\Delta : V \to V \otimes V$, the couple $(V, \Delta)$ is called a "weak co-UJLA structure" if this co-product satisfies the identity*

$$(Id + S + S^2) \circ (\Delta \otimes I) \circ \Delta = (Id + S + S^2) \circ (I \otimes \Delta) \circ \Delta \tag{8}$$

*where $S : V \otimes V \otimes V \to V \otimes V \otimes V$, $a \otimes b \otimes c \mapsto b \otimes c \otimes a$, $I : V \to V$, $a \mapsto a$ and $Id : V \otimes V \otimes V \to V \otimes V \otimes V$, $a \otimes b \otimes c \mapsto a \otimes b \otimes c$.*

**Definition 3.** *Given a vector space V, with a linear map $\phi : V \otimes V \to V \otimes V$, the couple $(V, \phi)$ is called a "weak (co)UJLA structure" if the map $\phi$ satisfies the identity*

$$(Id + S + S^2) \circ \phi^{12} \circ \phi^{23} \circ \phi^{12} \circ (Id + S + S^2) = (Id + S + S^2) \circ \phi^{23} \circ \phi^{12} \circ \phi^{23} \circ (Id + S + S^2) \tag{9}$$

*where $\phi^{12} = \phi \otimes I$, $\phi^{23} = I \otimes \phi$, $Id : V \otimes V \otimes V \to V \otimes V \otimes V$, $a \otimes b \otimes c \mapsto a \otimes b \otimes c$ and $I : V \to V$, $a \mapsto a$.*

**Theorem 1.** *Let $(V, \eta)$ be a weak UJLA structure with the unity $1 \in V$. Let $\phi : V \otimes V \to V \otimes V$, $a \otimes b \mapsto ab \otimes 1$. Then, $(V, \phi)$ is a "weak (co)UJLA structure".*

**Proof.** $(Id + S + S^2) \circ \phi^{23} \circ \phi^{12} \circ \phi^{23} \circ (Id + S + S^2)(a \otimes b \otimes c) = (Id + S + S^2) \circ \phi^{23} \circ \phi^{12} \circ \phi^{23}(a \otimes b \otimes c + b \otimes c \otimes a + c \otimes a \otimes b) = (Id + S + S^2) \circ \phi^{23} \circ \phi^{12}(a \otimes bc \otimes 1 + b \otimes ca \otimes 1 + c \otimes ab \otimes 1) = (Id + S + S^2) \circ \phi^{23}(a(bc) \otimes 1 \otimes 1 + b(ca) \otimes 1 \otimes 1 + c(ab) \otimes 1 \otimes 1) = (Id + S + S^2)(a(bc) \otimes 1 \otimes 1 + b(ca) \otimes 1 \otimes 1 + c(ab) \otimes 1 \otimes 1) = a(bc) \otimes 1 \otimes 1 + b(ca) \otimes 1 \otimes 1 + c(ab) \otimes 1 \otimes 1 + 1 \otimes 1 \otimes a(bc) + 1 \otimes 1 \otimes b(ca) + 1 \otimes 1 \otimes c(ab) + 1 \otimes a(bc) \otimes 1 + 1 \otimes b(ca) \otimes 1 + 1 \otimes c(ab) \otimes 1$.

Similarly,

$(Id + S + S^2) \circ \phi^{12} \circ \phi^{23} \circ \phi^{12} \circ (Id + S + S^2)(a \otimes b \otimes c) = (Id + S + S^2) \circ \phi^{12} \circ \phi^{23} \circ \phi^{12}(a \otimes b \otimes c + b \otimes c \otimes a + c \otimes a \otimes b) = (ab)c \otimes 1 \otimes 1 + (bc)a \otimes 1 \otimes 1 + (ca)b \otimes 1 \otimes 1 + 1 \otimes 1 \otimes (ab)c + 1 \otimes 1 \otimes (bc)a + 1 \otimes 1 \otimes (ca)b + 1 \otimes (ab)c \otimes 1 + 1 \otimes (bc)a \otimes 1 + 1 \otimes (ca)b \otimes 1$.

We now use the axiom of the "weak UJLA structure". $\square$

**Theorem 2.** *Let $(V, \Delta)$ be a weak co-UJLA structure with the co-unity $\varepsilon : V \to k$. Let $\phi = \Delta \otimes \varepsilon : V \otimes V \to V \otimes V$. Then, $(V, \phi)$ is a "weak (co)UJLA structure".*

**Proof.** The proof is dual to the above proof. We refer to [6–8] for a similar approach.

A direct proof should use the property of the co-unity: $(\varepsilon \otimes I) \circ \Delta = I = (I \otimes \varepsilon) \circ \Delta$. After computing

$$\phi^{12} \circ \phi^{23} \circ \phi^{12}(a \otimes b \otimes c) = \varepsilon(b)\varepsilon(c)(a_1)_1 \otimes (a_1)_2 \otimes a_2 \quad \text{and}$$
$$\phi^{23} \circ \phi^{12} \circ \phi^{23}(a \otimes b \otimes c) = \varepsilon(b)\varepsilon(c)a_1 \otimes (a_2)_1 \otimes (a_2)_2,$$

one just checks that the properties of the linear map $Id + S + S^2$ will help to obtain the desired result. □

**Theorem 3.** *Let $(V, \eta)$ be a weak UJLA structure with the unity $1 \in V$. Let $\phi : V \otimes V \to V \otimes V$, $a \otimes b \mapsto ab \otimes 1 + 1 \otimes ab - a \otimes b$. Then, $(V, \phi)$ is a "weak (co)UJLA structure".*

**Proof.** One can formulate a direct proof, similar to the proof of Theorem 1.

Alternatively, one could use the calculations from [7] and the axiom of the "weak UJLA structure". □

## 3. Unification of (Co)Derivations and Applications

**Definition 4.** *Given a vector space V, a linear map $d : V \to V$, and a linear map $\phi : V \otimes V \to V \otimes V$, with the properties*

$$\phi^{12} \circ \phi^{23} \circ \phi^{12} = \phi^{23} \circ \phi^{12} \circ \phi^{23} \tag{10}$$

$$\phi \circ \phi = Id, \tag{11}$$

*the triple $(V, d, \phi)$ is called a "generalized derivation" if the maps d and $\phi$ satisfy the identity*

$\phi \circ (d \otimes I + I \otimes d) = (d \otimes I + I \otimes d) \circ \phi$.

*Here, we have used our usual notation: $\phi^{12} = \phi \otimes I$, $\phi^{23} = I \otimes \phi$, $Id : V \otimes V \to V \otimes V$, $a \otimes b \mapsto a \otimes b$ and $I : V \to V$, $a \mapsto a$.*

**Theorem 4.** *If A is an associative algebra and $d : A \to A$ is a derivation, and $\phi : A \otimes A \to A \otimes A$, $a \otimes b \mapsto ab \otimes 1 + 1 \otimes ab - a \otimes b$, then $(A, d, \phi)$ is a "generalized derivation".*

**Proof.** According to [7], $\phi$ verifies conditions (10) and (11). Recall now that $\quad d(ab) = d(a)b + ad(b) \; \forall a, b \in A$, $\; d(1_A) = 0$.

$(d \otimes I + I \otimes d) \circ \phi(a \otimes b) = (d \otimes I + I \otimes d)(ab \otimes 1 + 1 \otimes ab - a \otimes b) = d(ab) \otimes 1 - d(a) \otimes b + 1 \otimes d(ab) - a \otimes d(b)$.

$\phi \circ (d \otimes I + I \otimes d)(a \otimes b) = \phi(d(a) \otimes b + a \otimes d(b)) = d(a)b \otimes 1 + 1 \otimes d(a)b - d(a) \otimes b + ad(b) \otimes 1 + 1 \otimes ad(b) - a \otimes d(b)$. □

**Theorem 5.** *If $(C, \Delta, \varepsilon)$ is a co-algebra, $d : C \to C$ is a co-derivation, and $\psi = \Delta \otimes \varepsilon + \varepsilon \otimes \Delta - Id : C \otimes C \to C \otimes C$, $c \otimes d \mapsto \varepsilon(d)c_1 \otimes c_2 + \varepsilon(c)d_1 \otimes d_2 - c \otimes d$, then $(C, d, \psi)$ is a "generalized derivation". (We use the sigma notation for co-algebras.)*

**Proof.** The proof is dual to the above proof.

According to [7], $\psi$ verifies conditions (10) and (11). From the definition of the co-derivation, we have $\varepsilon(d(c)) = 0$ and $\Delta(d(c)) = d(c_1) \otimes c_2 + c_1 \otimes d(c_2) \; \forall c \in C$.

$\psi \circ (d \otimes I + I \otimes d)(c \otimes a) = \varepsilon(a)d(c)_1 \otimes d(c)_2 - d(c) \otimes a + \varepsilon(c)d(a)_1 \otimes d(a)_2 - c \otimes d(a)$,

$(d \otimes I + I \otimes d) \circ \psi(c \otimes a) = \varepsilon(a)d(c_1) \otimes c_2 + \varepsilon(c)d(a_1) \otimes a_2 - d(c) \otimes a + \varepsilon(a)c_1 \otimes d(c_2) + \varepsilon(c)a_1 \otimes d(a_2) - c \otimes d(a)$.

The statement follows on from the main property of the co-derivative. □

**Definition 5.** *Given an associative algebra A with a derivation d : A → A, M an A-bimodule and D : M → M with the properties*

$$D(am) = d(a)m + aD(m) \quad D(ma) = D(m)a + md(a) \ \forall a \in A, \ \forall m \in M,$$

*the quadruple $(A, d, M, D)$ is called a "module derivation".*

**Remark 1.** *A "module derivation" is a module over an algebra with a derivation. It can be related to the co-variant derivative from differential geometry. Definition 5 also requires us to check that the formulas for D are well-defined.*

*Note that there are some similar constructions and results in [9] (see Theorems 1.27 and 1.40).*

**Theorem 6.** *In the above case, $A \oplus M$ becomes an algebra, and $\delta : A \oplus M \to A \oplus M$, $(a \oplus m) \mapsto (d(a) \oplus D(m))$ is a derivation of this algebra.*

**Proof.** We just need to check that $\delta((a \oplus m)(b \oplus n)) = \delta((ab \oplus an + mb)) = d(ab) \oplus D(an + mb)$

equals $\delta((a \oplus m)(b \oplus n)) = \delta((a \oplus m))(b \oplus n) + (a \oplus m)\delta(b \oplus n) = (d(a) \oplus D(m))(b \oplus n) + (a \oplus m)(d(b) \oplus D(n)) = (d(a)b \oplus d(a)n + D(m)b) + (ad(b) \oplus aD(n) + md(b))$. □

**Remark 2.** *A dual statement with a co-derivation and a co-module over that co-algebra can be given.*

**Remark 3.** *The above theorem leads to the unification of module derivation and co-module derivation.*

## 4. Modified Yang–Baxter Equation

For $A \in M_n(\mathbb{C})$ and $D \in M_n(\mathbb{C})$, a diagonal matrix, we propose the problem of finding $X \in M_n(\mathbb{C})$, such that

$$AXA + XAX = D . \tag{12}$$

This is an intermediate step to other "modified" versions of the Yang–Baxter equation (see, for example, [10]).

**Remark 4.** *Equation (12) is related to the problem of the three matrices. This problem is about the properties of the eigenvalues of the matrices A, B and C, where $A + B = C$. A good reference is the paper [11]. Note that if A is "small" then $D - AXA$ could be regarded as a deformation of D.*

**Remark 5.** *Equation (12) can be interpreted as a "generalized eigenvalue problem" (see, for example, [12]).*

**Remark 6.** *Equation (12) is a type of Yang–Baxter matrix equation (see, for example, [13,14]) if $D = O_n$ and $X = -Y$.*

**Remark 7.** *For $A \in M_2(\mathbb{C})$, a matrix with trace -1 and*

$$D = -\begin{pmatrix} det(A) & 0 \\ 0 & det(A) \end{pmatrix}, \tag{13}$$

*Equation (12) has the solution $X = I'$.*

**Remark 8.** *There are several methods to solve (12). For example, for $A^3 = I_n$, one could search for solutions of the following type: $X = \alpha I_n + \beta A + \gamma A^2$. Now, (12) implies that $(2\alpha\beta + \gamma^2 + \alpha)A^2 + (\alpha^2 + 2\beta\gamma + \gamma)A + (2\alpha\gamma + \beta^2 + \beta)I_n - D = 0$.*

*It can be shown that we can produce a large class of solutions in this way, if D is of a certain type.*

**Funding:** This research received no external funding.

**Acknowledgments:** I would like to thank Dan Timotin for the discussions and the reference on the problem of the three matrices. I also thank the editors and the referees.

**Conflicts of Interest:** The author declares no conflict of interest.

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
