# Peer review of "Unification Theories: New Results and Examples"

_axioms, doi:10.3390/axioms8020060_

Round 1

Reviewer 1 Report

In this note, the author extends previous work on the unification of mathematical notions. Specifically, and basing on the unified approach to Jordan, Lie and associative algebras developed in a previous note to the jorunal, the results are applied to derive a unification of derivations and coderivations. As an application, an intermediate step to generalized versions of the Yang-Baxter equation is proposed, giving some examples to this construction.

Up to some typographical errors (see ``associtive” instead of ``associative“ in the statements of Theorem 3.2 and Definition 3.4, the note is acceptable after a minimal revision. 

Author Response

Dear Reviewer,

Thank you very much for your comments.

I corrected the typographical errors: ``associtive” instead of ``associative“ in the statements of Theorem 3.2 and Definition 3.4,

I made other changes also.

Best wishes,

Author

Reviewer 2 Report

I suugest that the term "Euler's formula" in line 2 of the Inntroduction should be changed to

"Euler-De Moivre formula".

The English style of the definitions should be changed (some of them are even highly incorrect from the view point of English Grammar). I suggest the following style:

definition 2.1- ([3]) Given a vector space $V$ with a linear map $\eta: V\otimes V \to V$, the couple

$(V,\eta)$ will be called a weak UJLA structure if the product $ab := \eta(a \otimes b))$

satisfies the identity

$$(ab)c + (bc)a +(ca)b = a(bc) +b a(ca) + c(ab)  \eqno(7) .$$

All the Definitions should be revised in a similat manner.

"Accordind" should be changed 2 times to 'according".

Remark 4.1: "three matrix problem" (grammatically incorrect without explained context)

should be written in a different manner.

I think the use of "dual proofs" should be explained shortly at least at the first appearence in th proof of Theorem 2.5.

"Everithing follows" : this is a too loosel style -- write "All the statements follow etc" instead.

"There exists a direct proof" (in the proof of Theorem 2.6): give either a reference or say: "One can formulate a direct proof etc.".

Author Response

Dear Reviewer,

Thank you for your suggestions

I changed the term "Euler's formula" in line 2 of the Introduction  to "Euler-De Moivre formula".

I have the style of the definitions as suggested by the referee.

"Accordind" was  changed 2 times to 'according".

Remark 4.1: "three matrix problem" was changed to "the problem of the three matrices"

The text of this remark was improved.

 "dual proofs" was explained shortly at the first appearance in the proof of Theorem 2.5.

"Everithing follows" : was changed.

"There exists a direct proof" (in the proof of Theorem 2.6): was changed to: "One can formulate a direct proof ".

Thank you.

Best wishes.

Author

Reviewer 3 Report

See remarks.

Author Response

Dear Reviewer,

Thank you for your report and reference.

I have added Grinsberg's paper in the references section. I tried to find similar results in that paper. I briefly explained this in Remark 3.5:

"  A   'module derivation'  is a module over an algebra with a derivation. (It is a categorical point

of view.)

It can be related to the covariant derivative from Differential Geometry. 

The Definition 3.4 also requires to check that formulas for D are well-defined. 

Note some similar constructions and results in [9] (see the Theorems 1.27 and 1.40)."

Bordermann's paper (in our new References) is just one example of a  "modified" Yang-Baxter equation. Of course, one needs to use the relationship between the CYBE and QYBE. 

I also have added more references, which enhance the perspectives on possible applications and connections,

Best wishes,

Author